# Immune Checkpoint Inhibitors in Oral Cavity Squamous Cell Carcinoma and Oral Potentially Malignant Disorders: A Systematic Review

**DOI:** 10.3390/cancers12071937

**Published:** 2020-07-17

**Authors:** Omar Kujan, Bede van Schaijik, Camile S. Farah

**Affiliations:** 1UWA Dental School, The University of Western Australia, Nedlands, WA 6009, Australia; 22582886@student.uwa.edu.au; 2Australian Centre for Oral Oncology Research & Education, Nedlands, WA 6009, Australia; camile@oralmedpath.com.au; 3Oral, Maxillofacial and Dental Surgery, Fiona Stanley Hospital, Murdoch WA 6150, Australia; 4Genomics for Life, Herston QLD 4006, Australia

**Keywords:** immune checkpoint inhibitors, PD-1, PD-L1, oral squamous cell carcinoma, oral potentially malignant disorders

## Abstract

Cancers of the oral cavity cause significant cancer-related death worldwide. While survival rates have improved in recent years, new methods of treatment are being investigated to limit disease progression and to improve outcomes, particularly in oral cavity squamous cell carcinoma (OSCC) and oral potentially malignant disorders (OPMD). The emerging treatment modality of immunotherapy targets immune checkpoint molecules including PD-1 and its ligand PD-L1, CTLA-4, LAG-3, and TIM-3 to enhance the host immune response against tumours, and to limit the growth and progression of cancer cells. In this systematic review, we searched five databases for keywords pertaining to oral cancers and OPMDs, along with immune checkpoint inhibitors, in order to summarize the current status of their use and efficacy in these diseases. A total of 644 different articles were identified between 2004 and 2019, with 76 deemed suitable for inclusion in the study, providing a total of 8826 samples. Combined results show expression of PD-1 and PD-L1 in the majority of OPMD and OSCC samples, with expression correlating with increased progression and decreased survival rates. Immunotherapy agents pembrolizumab and nivolumab target PD-1 and have been shown to prolong survival rates and improve disease outcomes, especially in combination with chemotherapy or radiotherapy. Despite the equivocal nature of current evidence, there is support for the prognostic and predictive value of immune checkpoint molecules, especially PD-L1, and many studies provide support for the effective use of immune checkpoint inhibitors in the management of OSCC. Limited data is available for OPMD, therefore this should be the focus of future research.

## 1. Introduction

Cancer survival rates have improved significantly over the last few years [1]. In the treatment of head and neck malignancies, including oral cavity squamous cell carcinoma (OSCC), immune checkpoint inhibitors constitute a significant breakthrough influencing treatment outcomes and improving overall survival [1]. The use of monoclonal antibodies that block inhibitory immune checkpoint molecules, enhances the immune response to tumours, and consequently controls the growth and spread of neoplastic cells [2,3]. Immune checkpoint blockade removes inhibitory signals of T-cell activation, which enable tumour-reactive T cells to overcome regulatory mechanisms and mount an effective anti-tumour response [4]. The goal of cancer immunotherapy is to boost or restore the ability of the immune system to detect and destroy cancer cells. This is achieved by overcoming the mechanisms by which tumours evade and suppress the immune response, in essence, shifting the equilibrium back in favor of immune protection [5]. Regulatory T-cells play an important role in the tumour microenvironment (TME). They can mediate tolerance, suppress effector T-cells, and inhibit immune-mediated destruction. Amarnath et al. demonstrated that regulatory T cells (Tregs) cause dendritic cells to increase PD-L1 expression by affecting the PD-1–PD-L1 pathway [6]. PD-L1 expression by tumour cells can lead to evasion of the immune response by inhibiting T-cell responses through conversion of TH1 CD4 T-cells to Tregs [6]. This results in increased suppressive activity. PD-1/PD-L1 expression varies among different types of cancers and premalignant disorders. Immune checkpoint molecules include: PD-1, PD-L1, CTLA-4, TIM-3 and LAG-3. Targeted therapy with PD-1 or PD-L1 monoclonal antibodies shows durable tumour regression in patients with non-small cell lung cancer (NSCLC) who were non-responsive to chemotherapy or radiotherapy [7]. Phase I trials with anti-PD-1 or anti-PD-L1 have shown response rates of 18–28% and 10–17%, respectively, in patients with melanoma, NSCLC, and renal carcinoma [8]. PD-1 inhibitors pembrolizumab (Keytruda) and nivolumab (Opdivo) are used in the treatment of melanoma of the skin, NSCLC, kidney cancer, head and neck squamous cell carcinoma (HNSCC), and Hodgkins lymphoma, while PD-L1 inhibitors such as atezolizumab (Tecentriq) are used in the treatment of bladder cancer [9]. Clinical trials have demonstrated the utility of PD-1 and PD-L1 inhibitors in a wide variety of applications with relatively less toxic side effects compared to chemotherapy and radiotherapy [9]. Other immune checkpoint biomarkers that have shown promising results include CTLA-4, LAG-3, and TIM-3. When comparing PD-1/PD-L1 expression in HNSCC versus other epithelial cancers, one unique feature is its association with human papillomavirus (HPV), which has been shown to be the driver of carcinogenesis in 40–80% of oropharyngeal carcinomas, but not in malignancies arising from the oral cavity or other head and neck sites [10]. Recently the role of PD-1 and PD-L1 has been explored in premalignant lesions known collectively as oral potential malignant disorders (OPMD). In a group of patients with actinic cheilitis, the over-expression of PD-1/PD-L1 was higher than that of healthy volunteers but lower than that observed in OSCC [11]. This was investigated in order to identify the possibility of targeting immune checkpoint molecules prior to progression of OPMDs to OSCC. This systematic review aimed to unravel the expression of immune checkpoint biomarkers in the oral cavity subset of HNSCC, and to compare that in precursor lesions; OPMDs. A secondary aim was to identify biomarkers with prognostic or predictive value and to explore the role of therapeutic agents and their efficacy in the management of oral cavity SCC.

## 2. Materials and Methods

This systematic review was performed and reported according to the Preferred Reporting Items for Systematic Review and Meta-analysis (PRISMA) [12]. The article selection flow chart is presented in Figure 1.

### 2.1. PICO Statement

For this systematic review, the population of interest was patients with OSCC and OPMD of the oral cavity, the intervention was the use of immune checkpoint inhibitors, and the controls were normal healthy oral tissues. The outcome was to determine the effect of immune checkpoint inhibitors on the survival of patients with OSCC or OPMD.

### 2.2. Data Sources and Search Strategy

Electronic databases were searched by two authors (B.V.S. and O.K.). The databases searched were as follows: MEDLINE, EMBASE, PubMed, Web of Science and Scopus. The search strategies were developed using the specific (Medical Subject Headings) MeSH terms demonstrated in Table 1. References were checked from bibliographies in relevant articles and included in the systematic review if they were not identified initially. The search strategy included all studies published up to the end of December 2019.

### 2.3. Study Selection and Data Extraction

Initial studies identified through database searching were screened according to title and abstract against the inclusion criteria. The inclusion criteria were restricted to English language, immune checkpoint inhibitors as specified in the search statement, and lesions or tumours in the oral cavity. Review articles, case reports, non-English articles, animal studies or studies using cell lines were excluded. All studies considered eligible were included for full-text evaluation. When there was disagreement between the reviewers on whether to include or exclude a study, it was resolved through discussion. Then data were extracted from each included study using a standardised data collection form including lead author, publication year, study inclusion and exclusion criteria (when available), type of investigated specimens, sample size, subject breakdown (when available), subsites (when available), method of detection, and follow-up periods (when available), in addition to data on drug efficacy.

The main outcomes of this systematic review were to:

(1) Report on the expression of immune checkpoint biomarkers in OSCC and compare that to OPMD; (2) Establish whether the studied biomarkers had prognostic or predictive value; (3) Explore the role of recent therapeutic agents and their efficacy in the management of OSCC.

### 2.4. Quality Assessment

Evaluation of the studies included in the systematic review were assessed using the Quality Assessment of Diagnostic Accuracy Studies 2 (QUADAS-2) revised tool for the quality assessment of diagnostic accuracy studies [13], and Quality In Prognosis Studies (QUIPS) for prognostic and predictive studies [14].

## 3. Results

The initial search generated a total of 1756 articles from all databases used (PubMed 459, MEDLINE 318, EMBASE 391, Web of Science 346 and Scopus 242). After removing duplicates, a total of 644 articles were identified. These articles were assessed against the inclusion and exclusion criteria. In total, 433 articles were excluded after reading the abstract. A total of 212 articles were eligible for full-text assessment. An additional 136 articles were excluded after reading the full text as they did not meet the inclusion/exclusion criteria. A total of 76 studies met the criteria used in this systematic review (Appendix A). The year of publication ranged from 2004 to 2019. A total of 8826 samples were investigated; 8322 OSCC and 504 OPMD samples, respectively. The average age of all OSCC subjects was 61.2 years (range 15–96) and the male to female ratio was 2.8. The average age of all OPMD subjects was 50.17 years (range 8–105) and the male to female ratio was 1.388. A summary of the expression patterns, as well as the main findings from the included articles, are presented in Appendix A.

The most common methodology for assessing expression of immune checkpoint biomarkers was immunohistochemistry (IHC) in 38 studies [10,15,16,17,18,19,20,21,22,23,24,25,26,27,28,29,30,31,32,33,34,35,36,37,38,39,40,41,42,43,44,45,46,47,48,49,50,51], followed by flow cytometry (FC) and polymerase chain reaction (PCR) with 10 [52,53,54,55,56,57,58,59,60,61] and 8 [62,63,64,65,66,67,68,69] studies, respectively. A combination of both IHC and PCR were used in seven studies [70,71,72,73,74,75,76], both IHC and FC in four [11,77,78,79], and one study used a combination of IHC, PCR and FC [80]. In situ hybridization [81], immunoblotting [82], cell sorting [83] and genotyping [84] were used in a single study each, and four studies had alternative methodologies [85,86,87,88] such as computed tomography [85], tumor growth kinetics analysis [86], radiographic analysis [87], and observed treatment response [88].

PD-L1 was the most studied biomarker, analyzed in 53 studies [10,11,15,16,17,18,19,20,21,22,23,24,25,26,27,28,29,30,31,32,34,35,36,37,39,40,41,42,43,44,45,46,48,50,54,59,60,61,64,65,67,68,70,71,72,74,75,76,77,78,80,81,86], followed by PD-1 in 25 studies [11,15,24,26,35,38,43,47,49,51,53,56,58,59,61,66,73,74,77,78,79,80,81,84,86] and CTLA-4 in 16 studies [33,45,46,52,54,55,56,57,58,62,63,69,74,77,80,82]. Other biomarkers analyzed were TIM-3 and LAG-3 in nine [45,49,53,56,58,59,77,79,83] and four [45,49,56,78] studies, respectively.

Oral cavity, with 1249 samples, was the most reported site for OSCC. Other anatomical subsites specifically reported included oral tongue, floor of mouth, buccal mucosa, gingiva, alveolus, hard palate, lip, and retromolar trigone with 713, 275, 112, 111, 55, 44, 44, and 38 samples, respectively. A range of OPMD types were reported, the most common being OLP with 233 samples, followed by oral leukoplakia with 129, non-specified oral precancerous lesions with 120, and actinic cheilitis with 22 samples.

Quality assessment of included diagnostic accuracy studies was performed using the QUADAS-2 tool, which assessed bias domains of patient selection, index tests, reference standards, and overall flow (Table 2). Quality assessment of included predictive and prognostic studies was performed using the QUIPS tool, which assessed the risk of bias for participation, attrition, prognostic factor and outcome measurements, confounding, and statistical analysis (Table 3). Overall, the quality of the included papers was low, with poor quality of reference standards and a higher than expected level of bias. While this poses practical issues in terms of obtaining healthy samples, future studies should focus on comparing tumour data to appropriate controls, such as patient matched non-diseased oral mucosa, in order to more accurately gauge the effect of immune checkpoint inhibitors on tumour progression and patient survival.

## 4. Discussion

Recently, the role of immune checkpoint inhibitors in the TME has been extensively investigated [27,42,43,47,48,76,79]. In this systematic review, we have focused on the role of various immune checkpoint biomarkers and their inhibitors in OSCC and OPMDs.

The role of immune checkpoint inhibitors has also been investigated in OPMDs with the aim of identifying specific biomarkers that can predict OSCC transformation. PD-1 and its ligands were found to be expressed on infiltrating lymphocytes in oral lichen planus (OLP) which may have a role in tolerance induction in inflamed oral mucosa [51,71], however this expression was lower than that expressed on CD8+ infiltrating lymphocytes in early stage tongue OSCC [73]. In addition, a higher percentage of CD8+CD154+ and granzyme B+ cells were found in OLP patients [57]. Gene polymorphisms at position PD-1.3 (rs11568821) have been detected, but not associated with susceptibility to OLP [84]. In contrast, although PD-1 gene polymorphisms at positions PD-1.3 (rs11568821), PD-1.5 (rs2227981) and PD-1.9 (rs2227982) were not solely associated with susceptibility to HNSCCs, haplotype combinations emerging from these three loci may render susceptibility to HNSCCs [89]. The frequency of CD4+ and CD8+ T cells expressing PD-1 was higher in actinic cheilitis compared to normal tissues [11]. According to Yagyuu et al. [48] and Zhou et al. [61], dysplastic lesions expressing PD-L1 on epithelial and subepithelial cells can evade the immune host system, and inhibition of the PD-1/PD-L1 pathway can prevent malignant transformation in OPMDs [48,61]. PD-L1 expression in oral leukoplakia was closely associated with disease progression and CD8+ lymphocytes [22]. Bhosale et al. explored chromosomal alterations and gene expression changes associated with transformation of leukoplakia to OSCC and found that amplifications at 1p36.33 and 11q22.1 were strongly associated with poor clinical outcome [65]. The VISTA protein (V domain Ig suppressor of T cell activation) is more highly expressed in dysplastic tissues compared to normal [46].

The TME of OSCC supports adaptive immune responses that can be maintained through specific antigens [79]. Ngamphaiboon et al. reported that 83.9% of OSCC samples in their cohort (n = 203) showed positive expression of PD-L1 [36], while Takahashi et al. reported low expression in 40% and high expression in 60% of OSCC samples (n = 77) [42]. PD-1/PD-L1 expression in OSCC has also been associated with increased tumour-infiltrating lymphocytes (TILs) [60,68,70,81]. A positive correlation between PD-L1 expression and CD8+ and CD4+ TILs was found in tongue SCC, with higher CD4+ PD-1+ TILs compared to CD8+PD-1+ TILs [31]. CD4 expression was associated with poor prognosis in OSCC compared to CD25, which has a more favourable outcome [33], and combined expression of PD-1/PD-L1 significantly decreased the 5-year disease-specific survival rate [35]. High expression of PD-L1 was associated with poor clinical outcome [29,30]. In OSCC, the frequency of CD4+ and CD8+ T cells expressing PD-1 is higher than actinic cheilitis [11]. CTLA-4 pleomorphism was associated with OSCC age of onset and survival [69]. The role of Tregs in modifying the anti-tumour response has been assessed by Gasparoto et al., who found that they play a part in inhibiting T-cell proliferation and secreting immunosuppressive cytokines [52]. This was also demonstrated in a study by Aggarwal et al. [82]. A recent study suggests that young patients with OSCC have a more favourable outcome with PD-1 blockade when compared to older counterparts [72]. Higher incidence of PD-L1 expression has been reported in OSCC in females with an associated inflammatory phenotype [40]. Strong cytoplasmic expression of PD-L1 was found in circulating tumour cells (CTCs) and this plays a role in limiting T cell activity in inflammatory responses [75]. In a recent study, high endothelial venules were found to be markers for favourable antitumour immune microenvironments in OSCC [76]. Bharti et al. studied the different genotypes associated with CTLA-4 and showed that the AA genotype had a stronger association with an increased risk of developing OSCC compared to the GG phenotype [62].

PD-L1 expression is common in OSCC at both the transcriptional and protein levels [34,44,47,50,81]. PD-L1 expression was noted to be different between primary and metastatic sites [10,32]. In addition, PD-1, PD-L1, OX40, and CTLA-4 are overexpressed in CTCs [80]. An association between PD-L1 and p16^INK4A^ expression in non-oropharyngeal SCC has been demonstrated [20]. In addition, PD-L1 expression was positively associated with p16 and Ki-67 [20,38]. HPV-negative OSCC has higher numbers of CD8+ T cells expressing CTLA-4 compared to HPV-positive OSCC, with non-smokers showing a greater benefit of IDO1 and PD-1/PD-L1 blockade compared to smokers [27,45,58]. Rasmussen et al. reported that the variable intratumoural expression of PD-L1 limits its utility as a biomarker, however identifying the optimal cutoff point for discriminating between responders and non-responders to specific agents would be a useful mechanism for determining treatment efficacy [37].

The predictive and prognostic value of immune checkpoint biomarkers (Table 4) is examined throughout the included articles. Expression of PD-L1 on tumour cells is predictive of improved overall survival and decreased recurrence rates in patients treated with immunotherapy [48,53,72]. Fielder et al. showed an association between PD-L1 expression in tumour cells and radio-sensitivity, and favourable survival following radiotherapy [26], while Seiwert et al. showed that PD-L1 expression levels were predictive of the best overall response and progression-free survival, and that SCC tumours with greater PD-L1 expression by IHC show greater antitumour activity [41]. Groeger et al. [17] showed that the 5-year survival rate of patients whose tissues were positive for B7-H1 (PD-L1) expression was 73.33% (11 of 15), while Hanna et al. [72] showed that membranous PD-L1 expression was associated with a decreased risk of death among female patients, with a highly statistically significant (*p* < 0.001) hazard ratio of 0.58. Ferris et al. [25], however, showed that treatment with nivolumab (anti-PD-1) demonstrated favourable outcomes regardless of tumour cell PD-L1 expression, while Satgunaseelan et al. [40] presented no correlation with PD-L1 expression and overall survival in patients receiving post-operative adjuvant therapy, indicating that the efficacy of immune checkpoint inhibitors may also be independent of ligand expression. Conversely, Chen et al. showed that positive PD-L1 expression in OSCC with necrosis had worse survival outcomes and poorer disease control, however this is likely due to hypoxia of surrounding tissue overriding the effects of PD-L1 expression [21]. Moreira et al. found no association between CTLA-4 expression and survival rate in OSCC patients [33].

In terms of prognostic value, Malaspina et al. found that high PD-1 expression in CD4+ (43%) and CD8+ (68%) T cells may be used as a marker of poor prognosis in OSCC and in actinic cheilitis [11]. Furthermore, Naruse et al. determined that combined expression of PD-1 and PD-L1 significantly decreased the 5-year survival of OSCC patients [35]. Maruse et al. examined the 5-year survival of OSCC patients and found that PD-L1 positivity was associated with approximately 20% lower survival than PD-L1 negativity, however found no significant difference in 5-year survival between PD-1-positive and negative groups [30]. Both PD-L1 and PD-1 expression is significantly associated with cervical lymph node metastasis, with odds ratios of 3.67 and 3.99, respectively [30], and higher baseline levels of soluble PD-L1 correlated with nodal status, with higher expression levels in patients with node-positive disease [59]. Okada et al. showed that the 5-year overall survival rates in low and high PD-L1 groups were 72.5% and 16.7%, respectively [18], and Ngamphaiboon et al. showed that highly expressed PD-L1 (≥50%) was an independent prognostic factor for poor overall survival in anti-PD-1/PD-L1 untreated OSCC patients [36], further supporting PD-L1 as a prognostic biomarker of OSCC. These results indicate that expression of PD-L1, and possibly PD-1, are useful predictive biomarkers of disease response to immunotherapy and are potentially useful prognostic biomarkers for poor prognosis and disease progression and metastasis.

In studies using IHC, PD-L1 exhibited membranous and cytoplasmic expression in OSCC [34,72], with minimal nuclear staining [35,40], and higher expression was observed in OSCC compared to oral leukoplakia [22]. Cytoplasmic PD-L1 expression was histologically found in areas with poorly differentiated cells showing nuclear atypia [19,75]. PD-1 IHC expression was limited to lymphocytes infiltrating or surrounding tumour nests [35] with higher expression in OSCC compared to actinic cheilitis [11], while CTLA-4 was expressed in TILs within the TME in contrast to Tregs [46,73,80]. IHC analysis showed that TIM-3 was expressed at a higher rate in CD4 and CD8 T cells compared with matched normal blood samples [79], and treatment of OSCC with nimotuzumab appeared to increase the expression of PD-L1, CTLA-4, TIM-3 and LAG-3 [45].

Malignant transformation of OPMD is a significant contributor to poor disease outcome. From the included studies, Yagyuu et al. [48] was the only article to examine the association of malignant transformation of OPMD and the expression of immune checkpoint molecules. This study demonstrated that in oral precancerous lesions, subepithelial PD-L1-positive cell count and epithelial PD-L1 positivity were significantly associated with malignant transformation, and therefore are indicators of a poor prognosis of OPMD. Despite this, more evidence is required to comprehensively understand the link between immune checkpoint biomarkers and malignant transformation of OPMD.

Similarly, only two of the included studies compared OPMD to OSCC lesions, therefore it is difficult to compare the significance of immune checkpoint molecule expression between these disease states. Chen et al. 2019 [22] assessed PD-L1 expression between OSCC and OLK and determined that expression was greater in OSCC than OLK, suggesting that relative abundance of PD-L1 may be an indicator of disease progression. Furthermore, Malaspina et al. [11] demonstrated that OSCC tumour samples showed higher expression of PD-1 and higher numbers of CD4+ PD-1+ T cells when compared with tissue from actinic cheilitis patients, and in turn from healthy controls, further suggesting the possible capacity of expression to indicate disease severity and progression. Given that these studies were not longitudinal in nature, it is difficult to draw any meaningful conclusions in relation to malignant progression, hence further studies with stronger evidence are required to assess a possible relationship.

Therapeutic drug trials are still being conducted with the aim of delivering targeted therapies with less side effects. The role of immune checkpoint inhibitor chemotherapeutic treatment and PD-L1 expression was explored in OSCC [43,87]. Pembrolizumab (anti-PD-1) is FDA-approved for the management of patients with recurrent or metastatic OSCC [41,88]. It exhibits anti-tumour activity and an acceptable safety profile compared to cabazitaxel [16]. Falco et al. found that 22 out of 40 patients examined achieved clinical benefit from pembrolizumab—partial response seen in 10, stable disease 9, and complete response in 3 patients [87]. Nivolumab (anti-PD-1) was found to prolong survival rate compared to standard therapy with fewer toxic side effects [25]. The role of PD-1/PD-L1 immune checkpoint molecules are currently being investigated in relation to radiation therapy. PD-L1 indicate radio-sensitivity while survivin and c-Met implicates radio-resistance [26]. Moreover, fractionated chemoradiation leads to quantifiable effects and balance between suppressive and stimulatory mechanisms, which will be helpful when combined with immune checkpoint blockade [59]. Many genes are involved in tumorigenesis pathways, but of interest, six genes (*PGF, PD-L1/CD274, CDK6, EGFR, MET, VEGFA*) are overexpressed in HNSCC and associated with poorer outcome [39]. Furthermore, mPDCD1 (methylated PD-1 promotor) might potentially serve as a predictive biomarker for the response to immunotherapies targeting the PD-1/PD-L1 axis [66]. As noted by Feldman et al. in relation to HNSCC management, data supports the use of specific agents (PIK3CA, PD-1/PD-L1), combination strategies (PIK3CA plus EGFR) or agents approved for other solid tumours such as MGMT (O^6^-methylguanine DNA methyltransferase) or HER2 (human epidermal growth factor receptor 2) [24]. PD-1 methylation may aid in identifying HNSCC patients who might benefit from targeted immunotherapy against the PD-1/PD-L1 pathway [66]. Combination therapy with docetaxel, platinum, and fluorouracil modifies PD-L1 expression by increasing PD-L1 positivity [28]. This has great implications when trying to craft a therapeutic strategy.

Current literature on the use of immune checkpoint inhibitors in OSCC focuses on their efficacy in improving overall survival and progression-free survival. A recent systematic review by Ghanizada et al. examined the effect of immunotherapy on HNSCC, summarizing that checkpoint inhibition exhibits an anti-tumour effect [90]. Building on this study, our systematic review examines the predictive and prognostic value of immune checkpoint biomarkers, and also compares OSCC with OPMD. Similarly, another recent systematic review by Yang et al. examined the prognostic role of PD-L1 in HNSCC, and our analysis builds on this by examining other immune checkpoint biomarkers, as well as their predictive values [91]. Both of these articles broadly examine HNSCC, therefore our study fills a crucial gap in the specific examination of OSCC and the comparison to OPMD precursor lesions.

A limitation of our systematic review is the reported low quality of the included articles. Many of these articles have a high level of bias, therefore it is difficult to compare results between studies. Furthermore, the range of article types means that no single quality assessment tool was sufficient to grade them all, therefore it was necessary to utilize both QUADAS-2 and QUIPS to determine the quality of included articles. The low-quality of included articles highlights that future research should focus on improving study design and accounting for bias, as well as improving the adequacy and suitability of control tissue for comparison to both OPMD and OSCC. Comparison of OPMD and progression to OSCC in longitudinal patient cohorts is also required.

## 5. Conclusions

The studies included in this systematic review examined the expression of immune checkpoint molecules in human OSCC and OPMD. There is a significant gap in the literature pertaining to the comparative expression of these biomarkers to normal tissue and there is a need to standardize detection methods and the classification of positive expression across different tumour types and subsites within the oral cavity, in order to make the data less ambiguous. Despite the equivocal nature of current evidence, there is support for the prognostic and predictive values of immune checkpoint molecules, especially PD-L1, and many studies provide support for the effective use of immune checkpoint inhibitors in the management of OSCC. Limited data is available for OPMD, therefore this should be the focus of future research.

## Figures and Tables

**Figure 1 cancers-12-01937-f001:**
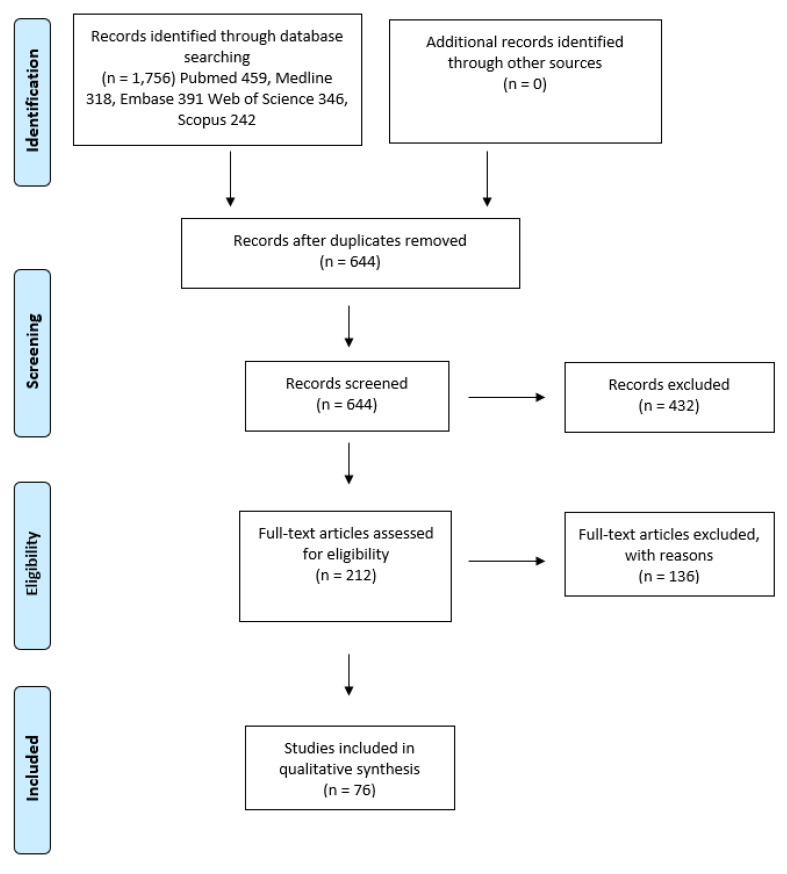
Article selection flow chart for the systematic review using Preferred Reporting Items for Systematic Review and Meta-analysis (PRISMA) guidelines.

**Table 1 cancers-12-01937-t001:** Medical Subject Headings (MeSH) Terms for the Search Strategy.

No.	Search Strategy
1	Oral or gingiva or buccal or tongue or oropharyngeal or cheek or lip or lingual or floor of the mouth or tonsils or retromolar or palate or mandible or maxilla
2	Squamous cell carcinoma or epithelial dysplasia or oral potentially malignant disorders or oral potentially malignant lesions or oral precancer or oral cancer or leukoplakia or lichen planus or erythroplakia or OSCC
3	PD-1 or PDL-1 or PD-1 or PDL1 or CD274 or CD279 or CTLA-4 or CTLA4 or LAG-3 or LAG3 or TIM-3 or TIM3 or CD152 or CD223 or immune checkpoint inhibitors

**Table 2 cancers-12-01937-t002:** Quality Assessment of Diagnostic Accuracy Studies Using the Diagnostic Accuracy Studies 2 QUADAS-2 Revised Tool.

Author/Year	Domain 1: Patient Selection		Domain 2: Index test(s)		Domain 3: Reference Standard		Domain 4: Flow and timing
	(A) Risk of Bias	(B) Concerns regarding applicability	(A) Risk of Bias	(B) Concerns regarding applicability	(A) Risk of Bias	(B) Concerns regarding applicability	Risk of Bias
Aggarwal et al. 2017 [82]	high	high	low	low	unclear	unclear	low
Bharti et al. 2013 [62]	low	low	low	low	low	low	low
Bhosale et al. 2017 [65]	low	low	low	low	low	low	low
Chen et al. 2018 [20]	high	low	high	unclear	low	low	low
Chen et al. 2019 [22]	low	low	low	low	low	low	low
Cho et al. 2011 [23]	low	low	low	low	low	low	low
Dong et al. 2017 [83]	low	unclear	unclear	unclear	unclear	unclear	low
Du et al. 2011 [71]	low	low	low	low	low	low	low
Fayette et al. 2017 [85]	low	high	low	low	low	low	high
Feldman et al. 2015 [24]	low	unclear	unclear	unclear	low	low	low
Ferris et al. 2017 [25]	low	low	low	low	low	low	high
Gasparoto et al. 2010 [52]	low	unclear	low	low	low	low	low
Ghapanchi et al. 2019 [84]	Low	Low	Low	Low	Low	Low	Low
Jie et al. 2013 [56]	low	low	low	low	low	low	low
Kämmerer et al. 2010 [63]	unclear	unclear	low	low	low	low	unclear
Katou et al. 2007 [73]	low	low	low	unclear	low	low	high
Larkins et al. 2017 [88]	high	high	unclear	unclear	low	low	high
Lechner et al. 2017 [80]	low	low	low	low	low	low	low
Leduc et al. 2017 [28]	low	low	unclear	unclear	low	low	unclear
Linedale et al. 2017 [77]	low	low	low	low	low	low	low
Malm et al. 2015 [78]	high	high	low	low	unclear	unclear	unclear
Mattox et al. 2017 [31]	low	low	low	unclear	low	low	low
Oliveira-Costa et al. 2015 [75]	low	low	low	low	low	low	low
Pekiner et al. 2012 [57]	unclear	low	low	low	low	low	low
Poropatich et al. 2017 [58]	low	low	low	low	low	low	low
Quan et al. 2016 [79]	low	unclear	low	low	low	low	low
Rasmussen et al. 2019 [37]	Low	Low	Low	Low	unclear	unclear	low
Saâda-Bouzid et al. 2017 [86]	low	low	low	low	high	unclear	low
Satgunaseelan et al. 2017 [40]	low	low	low	low	low	low	low
Scognamiglio et al. 2017 [10]	low	unclear	low	low	low	low	low
Singh et al. 2017 [64]	low	low	unclear	unclear	unclear	unclear	low
Strauss et al. 2017 [55]	unclear	unclear	low	low	unclear	unclear	high
Strati et al. 2017 [67]	low	low	low	low	low	low	low
Takahashi et al. 2019 [42]	low	low	low	low	unclear	unclear	unclear
Takakura et al. 2017 [43]	low	unclear	low	low	low	low	low
Wirsing et al. 2018 [76]	low	low	low	low	low	low	low
Wu et al. 2017 [46]	low	low	low	low	low	low	low
Xu et al. 2019 [47]	low	low	unclear	unclear	unclear	unclear	high
Yang et al. 2019 [49]	low	low	low	low	low	low	low
Youngnak-Piboonratanakit et al. 2004 [51]	low	low	high	low	low	low	low
Zhou et al. 2012 [61]	unclear	low	low	low	unclear	unclear	high

**Table 3 cancers-12-01937-t003:** Quality Assessment of Studies of Prognostic Factors Using the Quality In Prognosis Studies QUIPS Tool.

Author/Year	Risk of Bias for Each Domain					
	Study Participation	Study Attrition	Prognostic Factor Measurement	Outcome Measurement	Study Confounding	Statistical Analysis and Reporting
Ahn et al. 2017 [70]	Moderate	Low	Low	Low	Moderate	Low
Balermpas et al. 2017 [15]	Low	Low	Low	Low	Moderate	Low
Bauml et al. 2017 [16]	Moderate	Low	Low	Low	Low	Low
Cai et al. 2019 [19]	Low	Low	Low	Moderate	Moderate	Low
Chen et al. 2015 [21]	Moderate	Low	Low	Moderate	Moderate	Moderate
Falco et al. 2019 [87]	Low	Low	High	High	Moderate	High
Fiedler et al. 2017 [26]	Moderate	High	Low	Low	Low	High
Foy et al. 2017 [27]	Low	Low	Low	Low	Low	Low
Goltz et al. 2017 [66]	Moderate	Low	Moderate	Moderate	Moderate	Low
Groeger et al. 2016 [17]	High	Low	Low	Low	Low	Low
Hanna et al. 2017 [72]	Low	Low	Low	Low	Low	Low
Hanna et al. 2017 [53]	Low	High	Low	Low	Moderate	Low
Lecerf et al. 2019 [74]	Low	Low	Moderate	Low	Low	Low
Lin et al. 2015 [29]	Low	Low	Low	Low	Low	Low
Malaspina et al. 2011 [11]	Low	Low	Low	Moderate	Moderate	Low
Maruse et al. 2018 [30]	Moderate	Low	Low	Moderate	Moderate	Low
Moratin et al. 2019 [32]	Moderate	Low	Low	Low	Moderate	Low
Moreira et al. 2010 [33]	Low	Moderate	Low	Moderate	Low	Low
Muller et al. 2017 [34]	Low	Low	Low	Moderate	Low	Low
Naruse et al. 2019 [35]	Moderate	Low	Low	Moderate	High	Moderate
Ngamphaiboon et al. 2019 [36]	Low	Low	Moderate	Moderate	Moderate	Moderate
Okada et al. 2018 [18]	High	Moderate	Low	Low	Low	Low
Ryu et al. 2017 [38]	Low	Moderate	Moderate	Moderate	High	Moderate
Sablin et al. 2016 [39]	Moderate	Low	Moderate	High	High	High
Seiwert et al. 2016 [41]	Low	Moderate	Moderate	High	Moderate	Low
Shayan et al. 2017 [54]	High	Low	Low	Low	Low	Low
Sridharan et al. 2016 [59]	High	Low	Moderate	Low	Moderate	Low
Straub et al. 2016 [81]	Low	Low	Low	Low	Low	Low
Troeltzsch et al. 2017 [44]	Low	Low	Low	Low	Moderate	Low
Wang et al. 2019 [45]	Moderate	Low	Moderate	Moderate	High	Low
Weber er al. 2018 [68]	Low	Moderate	Low	Low	Moderate	Low
Wong et al. 2006 [69]	Low	High	Low	Low	Low	Low
Wu et al. 2019 [60]	High	Moderate	Low	Low	High	High
Yagyuu et al. 2017 [48]	Low	Low	Low	Low	Low	Low
Yoo et al. 2019 [50]	Moderate	Moderate	Low	Moderate	Moderate	Low

**Table 4 cancers-12-01937-t004:** Predictive and Prognostic Values of Immune Checkpoint Biomarkers in Relevant Included Studies.

Author/Year	Biomarker	Tumour Type	Predictive Value	Prognostic Value
Ahn et al. 2017 [70]	PD-L1	OSCC	N/A	High PD-L1 expression was a favourable prognostic factor for overall survival only in the miR-197 high subgroup
Balermpas et al. 2017 [15]	PD-1, PD-L1	OSCC	N/A	PD-L1 in CD8 cells represents a promising prognostic marker and could be used to guide treatment with PD-1/PD-L1 inhibitors
Bauml et al. 2017 [16]	PD-L1	OSCC	17% of patients with PD-L1 expression of >1% of tumour cell sresponded to treatment with nivolumab	N/A
Cai et al. 2019 [19]	PD-L1	OSCC	Anti-PD-1 mAb be more efficacious in poorly differentiated OSCCs with higher PD-L1 expression	N/A
Chen et al. 2015 [21]	PD-L1	OSCC	OSCC patients with positive tumour PD-L1 expression may be good candidates for anti-PD-L1 immunotherapy	OSCC patients with necrosis and positive PD-L1 expression had worse outcomes and disease control
Falco et al. 2019 [87]	PD-L1	OSCC	In patients with high PD-L1 expression, single-agent pembrolizumab also improves overall survival compared with cetuximab plus chemotherapy	N/A
Fiedler et al. 2017 [26]	PD-1, PD-L1	OSCC	PD-L1 expression indicates radiosensitivity and favourable survival	N/A
Foy et al. 2017 [27]	PD-L1	OSCC	Overexpression of PD-L1 correlated with greater response to pembrolizumab	N/A
Goltz et al. 2017 [66]	PD-1	OSCC	mPDCD1 is a potential predictive biomarker for immunotherapies targeting the PD-1/PD-L1 axis	mPDCD1 high was associated with shorter overall survival
Groeger et al. 2016 [17]	PD-L1	OSCC	N/A	Expression of PD-L1 may be a prognostic marker for OSCC
Hanna et al. 2017 [72]	PD-L1	OSCC	N/A	Greater membranous PD-L1 positivity and the presence of TILs showed a decreased risk of recurrence and improved survival, hazard ratio 0.58
Hanna et al. 2017 [53]	PD-1, Tim3	OSCC	OSCC with inflamed immunophenotype benefit from single agent PD-1 blockade	N/A
Lecerf et al. 2019 [74]	PD-1, PD-L1, CTLA-4	OSCC	N/A	PD-1 overexpression was associated with good prognosis and low mRNA levels of PD-1 correlated with poor prognosis and high risk of recurrence
Lin et al. 2015 [29]	PD-L1	OSCC	N/A	High PD-L1-expression was significantly associated with distant metastasis and poor prognosis in male patients and smoking patients
Malaspina et al. 2011 [11]	PD-1, PD-L1	OSCC, Actinic Chelitis	N/A	High PD-1 expression in CD4+ (43%) and CD8+ (68%) T cells may be used as a potential prognostic marker in oral tumours or in pre-malignant lesions
Maruse et al. 2018 [30]	PD-1, PD-L1	OSCC	N/A	Co-expression of PD-L1 and PD-1 is predictive of a poor prognosis in OSCC patients
Moratin et al. 2019 [32]	PD-L1	OSCC	AntiPD-1/PD-L1 therapy may be of therapeutic use in early stage OSCC to prevent disease progression	There was a trend toward worse overall survival for patients with higher PD-L1 expression
Moreira et al. 2010 [33]	CTLA4	OSCC	N/A	No association between CTLA-4 expression and survival
Muller et al. 2017 [34]	PD-L1	OSCC	N/A	PD-L1 expression is a suitable biomarker for poor prognosis
Naruse et al. 2019 [35]	PD-1, PD-L1	OSCC	Inhibition of PD-1/PD-L1 axis may improve outcomes after non-adjuvant chemotherapy	Patients with combined PD-1+/PD-L1+ expressions had decreased 5-year disease-specific survival rate in oral tongue SCC
Ngamphaiboon et al. 2019 [36]	PD-L1	OSCC	N/A	PD-L1 expression above 50% indicated poor overall survival
Okada et al. 2018 [18]	PD-L1	OSCC	N/A	Low PD-L1 group had a better overall survival rate than high PD-L1 group (72.5% vs 16.7%)
Ryu et al. 2017 [38]	PD-1	OSCC	N/A	High PD-1+ T cells indicate worse prognosis
Sablin et al. 2016 [39]	PD-L1	OSCC	N/A	Overexpression of PD-L1 gene associated with poor outcome
Seiwert et al. 2016 [41]	PD-L1	OSCC	PD-L1 expression by IHC was predictive of best overall response and improved progression-free survival	N/A
Shayan et al. 2017 [54]	PD-L1, CTLA-4	OSCC	Addition of a PD-1 inhibitor to cetuximab and motolimod increases the antitumor response	N/A
Sridharan et al. 2016 [59]	PD-1, Tim3, PD-L1	OSCC	N/A	Higher baseline levels of soluble PD-L1 correlated with nodal status
Straub et al. 2016 [81]	PD-1, PD-L1	OSCC	N/A	PD-L1 positivity indicates increased risk of nodal metastasis, recurrence and death
Troeltzsch et al. 2017 [44]	PD-L1	OSCC	N/A	PD-L1 expression is associated with increased metastasis
Wang et al. 2019 [45]	PD-L1, TIM3, CTLA4, Lag-3	OSCC	N/A	Aberrant LAG-3 and PD-L1 expression was associated with worse survival
Weber et al. 2018 [68]	PD-L1	OSCC	N/A	Peripheral blood PD-L1 expression indicates metastatic disease
Wong et al. 2006 [69]	CTLA4	OSCC	N/A	CTLA-4 A/A genotype polymorphism is associated with poorer survival
Wu et al. 2019 [60]	PD-1	OSCC	Blockage of PD-1 and TIGIT elicits better anti-tumour effects	N/A
Yagyuu et al. 2017 [48]	PD-L1	Oral Precancerous Lesions	N/A	Subepithelial PD-L1-positive cell count and epithelial PD-L1 positivity were significantly associated with malignant transformation
Yoo et al. 2019 [50]	PD-L1	OSCC	N/A	Loss of MHC class I expression is significantly associated with a worse prognosis in PD-L1-positive OCSCC (hazard ratio = 4.24)

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
