# Peer review of "Immune Checkpoint Inhibitors in Oral Cavity Squamous Cell Carcinoma and Oral Potentially Malignant Disorders: A Systematic Review"

_cancers, 2020, doi:10.3390/cancers12071937_

Round 1
Reviewer 1 Report
I congratulate with the Authors for the extensive and well conducted analisys.
The article is well written and the data are clear and easily readable.
The topic is of interest and the results provide updated information for the scientific community.
The only concern I have with this paper is about table 1. It is too big, difficult to read and it should be simplified in order to make it easy to understand for the readers. I recommend to change the format (maybe horizontal instead of vertical layout).
Globally, the article could be published after the revision of table 1.
Author Response
We welcome and thank the reviewer for the comments and kind words. We have amended Table 1 by splitting it into 2 tables. Table 1 includes the extracted data from the inclusion criteria. Table 2 presents the patterns of expression and main findings. We trust that this shortens the manuscript length and makes it more presentable and readable, while maintaining all data crucial to our article.
Reviewer 2 Report
This paper is the study about “Immune checkpoint inhibitors in oral cavity squamous cell carcinoma and oral potentially malignant disorders: A systematic review”. We think that this study is interesting. However, the concept and significance are unclear as systematic review. Please make the purpose of this study clear and consider reconsideration.Some comments are as below:
- HPV is also associated with the oral cavity. Is this sentence meaningful in the first place?
- Table 1 is difficult to see. A radical correction is needed.
- Isn't Table 2, 3 a result? Need to revise the whole system.
- A systematic review should essentially extract high quality (sample size/RCT) studies. This research cannot be said to be of high quality.
- Should be compared non-cancerous cases and cancerous cases on OPMD.
- The purpose of OSCC and OPMD has not been compared, and the purpose is unknown.
Author Response
Thank you for your comments. Here is below our response:
- HPV is also associated with the oral cavity. Is this sentence meaningful in the first place?
Authors’ response: The current evidence indicates that there is a very little role of HPV in inducing oral cavity SCC. Only the strong association is found with oropharyngeal SCC. This sentence is relevant to the introduction to show the significant relationship between HPV and PD-1 in specific types of tumours.
- Table 1 is difficult to see. A radical correction is needed.
Authors’ response: We have amended Table 1 by splitting it into 2 tables. Table 1 includes the extracted data from the inclusion criteria. Table 2 presents the patterns of expression and main findings. We trust that this shortens the manuscript length and makes it more presentable and readable, while maintaining all data crucial to our article.
- Isn't Table 2, 3 a result? Need to revise the whole system.
Authors’ response: We have moved the quality assessment paragraphs to the results section as well as their corresponding tables.
- A systematic review should essentially extract high quality (sample size/RCT) studies. This research cannot be said to be of high quality.
Authors’ response: Systematic review reports what is available in the literature and it provides assessment of the methodological quality of included studies. This is a standard perquisite set by international reporting guidelines (PRISMA). Having said that, our study has used a stringent quality assessment tool and highlighted the low quality of included articles. We also provided suggestion for the need of higher quality studies as a future direction.
- Should be compared non-cancerous cases and cancerous cases on OPMD.
Authors’ response: OPMD may progress into cancer, but this only occurs in a proportion of lesions. Additionally, it is difficult to predict which OPMD will undergo malignant transformation. This is a field of great interest in the literature with no answers forthcoming currently. In our paper we reported on the prognostic value of studied biomarkers.
- The purpose of OSCC and OPMD has not been compared, and the purpose is unknown.
Authors’ response: It is unclear what the reviewer means by “purpose”. OPMD and OSCC are two different entities.
Reviewer 3 Report
OVERALL RECOMMENDATION
Accept after Major Revisions
RATING MANUSCRIPT
Originality/Novelty
Is the question original and well defined? Yes
Do the results provide an advance in current knowledge? Yes.
Significance
Are the results interpreted appropriately? Yes.
Are they significant? Probably the authors should conduct a meta-analysis.
Are all conclusions justified and supported by the results? Yes.
Are hypotheses and speculations carefully identified as such? Yes.
Quality of Presentation
Is the article written in an appropriate way? Yes, the Authors follow the PRISMA guidelines to write the article. Nevertheless, the authors should add the PRISMA checklist.
Are the data and analyses presented appropriately? Yes, it appears so.
Are the highest standards for presentation of the results used? Yes.
Scientific Soundness
is the study correctly designed and technically sound? Somehow. Probably, more stringent inclusion criteria should be used in order to have article similar each other.
Are the analyses performed with the highest technical standards? The authors conduct a systematic revision but they did not perform any meta-analysis. Meta-analysis should give major scientific evidence and support for the data presented.
Are the data robust enough to draw the conclusions? Yes.
Are the methods, tools, software, and reagents described with sufficient details to allow another researcher to reproduce the results? Yes.
Interest to the Readers
Are the conclusions interesting for the readership of the Journal? Yes
Will the paper attract a wide readership, or be of interest only to a limited number of people? The study is really readable as it tries to clarify issue and gives some advice for future research.
Overall Merit
Is there an overall benefit to publishing this work? Yes.
Does the work provide an advance towards the current knowledge? Yes.
Do the authors have addressed an important long-standing question with smart experiments? Yes.
English Level
Is the English language appropriate and understandable? Yes.
REVIEW REPORT
Brief summary
It was a pleasure to conduct the review of the work entitled “Immune checkpoint inhibitors in oral cavity squamous cell carcinoma and oral potentially malignant disorders: A systematic review”, in which the Authors conducted a systematic review of the available literature regarding the expression of biomarkers in oral cancer and oral potentially malignant disorders, and its prognostic, predictive and therapeutic value.
The authors conducted a systematic review about a very interesting and current topic for readers, providing juicy suggestion for future research. To this regard, a previous systematic review “The effects of checkpoint inhibition on head and neck squamous cell T carcinoma: A systematic review” was published by Ghanizada M et al. (2019) and it referred of a broader cancer population. Nevertheless, some major changes are needed to make it suitable for publication.
Below, you can find some considerations about the paper.
Broad comments
- Article Title
Is the title of the manuscript brief, appropriate, and indicative of the material which is contained in the manuscript? Yes.
- Abstract
Is the abstract concise? Yes.
Does it adequately describe the study? Yes.
Are the results and significances adequately presented? Somehow. The authors conduct a systematic revision but they did not perform any meta-analysis. Meta-analysis should give major scientific evidence and support for the data presented.
- Introduction: Review of the Literature
Has the author cited the pertinent, but only the pertinent, literature? Yes, it appears so.
Is the length of the introduction and the literature review appropriate or excessive? It is appropriate.
- Statement of Objectives
Is there a clear statement of the objectives of the study? Yes.
Are the objectives justified by the introduction? Yes.
- Description of Study Design: Material and Methods
Are the methods used in the study scientifically valid and technically correct? Yes.
Is the experimental group appropriate? There is no experimental group being a systematic review.
Are the procedures described in sufficient detail for a clear understanding? Yes. The procedures are described in a clear manner.
For studies of patient treatment, is the duration of the follow-up sufficiently long, and were the appropriate parameters of success or failure examined? -
Were the outcome criteria identified and were they objectively and reliably measured? The report of the revision of the available literature was well-conducted.
- Statistical Analysis
Were the methods of statistical analysis appropriate for the study? Somehow. The authors conduct a systematic revision but they did not perform any meta-analysis. Meta-analysis should give major scientific evidence and support for the data presented.
Did the author appropriately interpret significance and non-significance correctly? Yes.
Is review by a statistician needed? Yes or No? No.
- Results
Are the results and data gathered in the study presented in a clear and logical method? Yes.
Are tables and figures used to illustrate the data? Yes. Authors should add the PRISMA checklist.
- Discussion
Does the author explain the importance of his findings? Yes, in a clear way.
Are the results of this study discussed in light of other studies? Yes.
Are the comparisons with other studies appropriate and insightful? Not at all. A previous systematic review “The effects of checkpoint inhibition on head and neck squamous cell T carcinoma: A systematic review” was published by Ghanizada M et al. (2019) and it referred of a broader cancer population. Could the authors comment that systematic review about any difference or similarities?
- Conclusions
Are the conclusions consistent with the data and results presented in the manuscript? Yes.
Are the conclusions warranted by the results? Yes.
Are the conclusions overstated, too broad, or inappropriate, based on the data presented? No.
- Figures
Are the figures appropriate in number and clarity? Yes.
Should any of the figures be deleted or revised? No.
Are they appropriately cropped and labeled? Yes.
Do the illustrations need to be printed in color? No.
- References
Are the references current and accurate? Yes.
Are important references omitted? Not completly. Authors should check some recent and pertinent literature.
If excessive, which should be deleted? None.
- Grammar and Style
This article has not significant errors and its interpretation is easy to achieve.
- General Comments
Some major changes are needed to make it suitable for publication.
Specific comments
No specific comment referring to line numbers, tables or figures should be made.
Author Response
Reviewers report 3
Thank you for your comments. Here is below our response:
RATING MANUSCRIPT
Originality/Novelty
Is the question original and well defined? Yes
Do the results provide an advance in current knowledge? Yes.
Significance
Are the results interpreted appropriately? Yes.
Are they significant? Probably the authors should conduct a meta-analysis.
Authors’ response:
We considered undertaking a meta-analysis but because of the high heterogeneity of investigated samples in the included studies, meta-analysis became meaningless.
Are all conclusions justified and supported by the results? Yes.
Are hypotheses and speculations carefully identified as such? Yes.
Quality of Presentation
Is the article written in an appropriate way? Yes, the Authors follow the PRISMA guidelines to write the article. Nevertheless, the authors should add the PRISMA checklist.
Authors’ response:
We followed PRISMA statement and Journal guidelines in reporting our study. However, a PRISMA checklist is attached for your reference.
Are the data and analyses presented appropriately? Yes, it appears so.
Are the highest standards for presentation of the results used? Yes.
Scientific Soundness
is the study correctly designed and technically sound? Somehow. Probably, more stringent inclusion criteria should be used in order to have article similar each other.
Are the analyses performed with the highest technical standards? The authors conduct a systematic revision but they did not perform any meta-analysis. Meta-analysis should give major scientific evidence and support for the data presented.
Authors’ response:
We covered this point in our response above.
Are the data robust enough to draw the conclusions? Yes.
Are the methods, tools, software, and reagents described with sufficient details to allow another researcher to reproduce the results? Yes.
Interest to the Readers
Are the conclusions interesting for the readership of the Journal? Yes
Will the paper attract a wide readership, or be of interest only to a limited number of people? The study is really readable as it tries to clarify issue and gives some advice for future research.
Authors’ response:
We welcome your comments and we thank you.
Overall Merit
Is there an overall benefit to publishing this work? Yes.
Does the work provide an advance towards the current knowledge? Yes.
Do the authors have addressed an important long-standing question with smart experiments? Yes.
English Level
Is the English language appropriate and understandable? Yes.
REVIEW REPORT
Brief summary
It was a pleasure to conduct the review of the work entitled “Immune checkpoint inhibitors in oral cavity squamous cell carcinoma and oral potentially malignant disorders: A systematic review”, in which the Authors conducted a systematic review of the available literature regarding the expression of biomarkers in oral cancer and oral potentially malignant disorders, and its prognostic, predictive and therapeutic value.
The authors conducted a systematic review about a very interesting and current topic for readers, providing juicy suggestion for future research. To this regard, a previous systematic review “The effects of checkpoint inhibition on head and neck squamous cell T carcinoma: A systematic review” was published by Ghanizada M et al. (2019) and it referred of a broader cancer population. Nevertheless, some major changes are needed to make it suitable for publication.
Below, you can find some considerations about the paper.
Broad comments
- Article Title
Is the title of the manuscript brief, appropriate, and indicative of the material which is contained in the manuscript? Yes.
- Abstract
Is the abstract concise? Yes.
Does it adequately describe the study? Yes.
Are the results and significances adequately presented? Somehow. The authors conduct a systematic revision but they did not perform any meta-analysis. Meta-analysis should give major scientific evidence and support for the data presented.
- Introduction: Review of the Literature
Has the author cited the pertinent, but only the pertinent, literature? Yes, it appears so.
Is the length of the introduction and the literature review appropriate or excessive? It is appropriate.
- Statement of Objectives
Is there a clear statement of the objectives of the study? Yes.
Are the objectives justified by the introduction? Yes.
- Description of Study Design: Material and Methods
Are the methods used in the study scientifically valid and technically correct? Yes.
Is the experimental group appropriate? There is no experimental group being a systematic review.
Are the procedures described in sufficient detail for a clear understanding? Yes. The procedures are described in a clear manner.
For studies of patient treatment, is the duration of the follow-up sufficiently long, and were the appropriate parameters of success or failure examined? -
Were the outcome criteria identified and were they objectively and reliably measured? The report of the revision of the available literature was well-conducted.
- Statistical Analysis
Were the methods of statistical analysis appropriate for the study? Somehow. The authors conduct a systematic revision but they did not perform any meta-analysis. Meta-analysis should give major scientific evidence and support for the data presented.
Did the author appropriately interpret significance and non-significance correctly? Yes.
Is review by a statistician needed? Yes or No? No.
- Results
Are the results and data gathered in the study presented in a clear and logical method? Yes.
Are tables and figures used to illustrate the data? Yes. Authors should add the PRISMA checklist.
Authors’ response:
We covered this point in our response above.
- Discussion
Does the author explain the importance of his findings? Yes, in a clear way.
Are the results of this study discussed in light of other studies? Yes.
Are the comparisons with other studies appropriate and insightful? Not at all. A previous systematic review “The effects of checkpoint inhibition on head and neck squamous cell T carcinoma: A systematic review” was published by Ghanizada M et al. (2019) and it referred of a broader cancer
Authors’ response:
Thanks you for this valid point. We have cited the Ghanizada M et al 2019 article and included a comparison to this article in the discussion section.
- Conclusions
Are the conclusions consistent with the data and results presented in the manuscript? Yes.
Are the conclusions warranted by the results? Yes.
Are the conclusions overstated, too broad, or inappropriate, based on the data presented? No.
- Figures
Are the figures appropriate in number and clarity? Yes.
Should any of the figures be deleted or revised? No.
Are they appropriately cropped and labeled? Yes.
Do the illustrations need to be printed in color? No.
- References
Are the references current and accurate? Yes.
Are important references omitted? Not completly. Authors should check some recent and pertinent literature.
Authors’ response:
Thanks you for this valid point. We have cited the Ghanizada M et al 2019 article and included a comparison to this article in the discussion section.
If excessive, which should be deleted? None.
Round 2
Reviewer 2 Report
This paper is the study about “Immune checkpoint inhibitors in oral cavity squamous cell carcinoma and oral potentially malignant disorders: A systematic review”. We think that this study is interesting. However, the concept and significance are unclear as systematic review. Please make the purpose of this study clear and consider reconsideration.
Some comments are as below:
- Should be compared non-cancerous cases and cancerous cases on OPMD.
- The purpose of your paper is “to unravel the expression of immune checkpoint biomarkers in the oral cavity subset of HNSCC, and to compare that in OPMDs”?
Your results were only a list of papers and no comparisons are made.
- Materials and methods are written in the result. Need to revise the whole system again.
- The most important thing in OPMD is malignant transformation, and its biomarkers are important. Whether it can be extracted at the immune checkpoint is an issue. If you don't touch that point, the debate about OPMD is less meaningful. Please reconsider.
Author Response
We thank reviewer 3 for their time and comments.
1- Should be compared non-cancerous cases and cancerous cases on OPMD.
Authors response: I assume reviewer refers cancerours OPMD to those have shown malignant transformation. Using such a term is inconsistent with the published literature particularly the updated WHO classification of OPMD paper which is due to be published very soon. I co-authored the updated classification and the use of cancerous vs non-cancerous is not supported. However, a comparison of studies with valid malignant transformation data was carried out and the following paragraph was added.
Malignant transformation of OPMD is a significant contributor to poor disease outcome. From the included studies, Yagyuu et al. [48] was the only article to examine the association of malignant transformation of OPMD and the expression of immune checkpoint molecules. This study demonstrated that in oral precancerous lesions, subepithelial PD-L1-positive cell count and epithelial PD-L1 positivity were significantly associated with malignant transformation, therefore are indicators of a poor prognosis of OPMD. Despite this, more evidence is required to comprehensively understand the link between immune checkpoint biomarkers and malignant transformation of OPMD.
2- The purpose of your paper is “to unravel the expression of immune checkpoint biomarkers in the oral cavity subset of HNSCC, and to compare that in OPMDs”?
Your results were only a list of papers and no comparisons are made.
Authors response: The discussion section has been updated with the comparison between OPMD and OSCC.
3- Materials and methods are written in the result. Need to revise the whole system again.
Authors response: This is a systematic review that was conducted and reported using the PRISMA statement. Therefore, the results section is distinctive from methods and no additional changes can be made.
4- The most important thing in OPMD is malignant transformation, and its biomarkers are important. Whether it can be extracted at the immune checkpoint is an issue. If you don't touch that point, the debate about OPMD is less meaningful. Please reconsider.
Authors response: The discussion has been updated on malignant transformation. However, the available data is very scant.
Reviewer 3 Report
The changes are satisfactory to previous comments. The article should be published in the present form, nevertheless a careful check of typing errors is needed (i.e. table 5 caption).
Author Response
Thank you for your comments. Careful language check was done to correct any typos.